# Development and Evaluation of a Digital Health Intervention to Prevent Type 2 Diabetes in Primary Care: The PREDIABETEXT Study Protocol for a Randomised Clinical Trial

**DOI:** 10.3390/ijerph192214706

**Published:** 2022-11-09

**Authors:** Aina M. Galmes-Panades, Escarlata Angullo, Sofía Mira-Martínez, Miquel Bennasar-Veny, Rocío Zamanillo-Campos, Rocío Gómez-Juanes, Jadwiga Konieczna, Rafael Jiménez, Maria Jesús Serrano-Ripoll, Maria Antonia Fiol-deRoque, Jerónima Miralles, Aina M. Yañez, Dora Romaguera, Maria Clara Vidal-Thomas, Joan Llobera-Canaves, Mauro García-Toro, Catalina Vicens, Elena Gervilla-García, José Iván Oña, Narges Malih, Alfonso Leiva, Oana Bulilete, Juan José Montaño, Margalida Gili, Miquel Roca, Ignacio Ricci-Cabello

**Affiliations:** 1Global Health and Lifestyle (EVES Group), Health Research Institute of the Balearic Islands (IdISBa), University Hospital Son Espases (HUSE), 07120 Palma, Spain; 2Consorcio CIBER, M.P. Fisiopatología de la Obesidad y Nutrición (CIBERObn), Instituto de Salud Carlos III (ISCIII), 28029 Madrid, Spain; 3Department of Nursing and Physiotherapy, University of the Balearic Islands (UIB), 07120 Palma, Spain; 4Escola Graduada Primary Health Care Center, Balearic Health Service, 07002 Palma de Mallorca, Spain; 5Research Group on Community Nutrition & Oxidative Stress, University of Balearic Islands-IUNICS, IDISBA & CIBEROBN (Physiopathology of Obesity and Nutrition), 07120 Palma de Mallorca, Spain; 6Health Research Institute of the Balearic Islands (IdISBa), 07120 Palma de Mallorca, Spain; 7Research Group in Primary Care and Promotion—Balearic Islands Community (GRAPP-caIB), 07120 Palma de Mallorca, Spain; 8Primary Care Research Unit of Mallorca (IB-Salut), Balearic Health Service, 07002 Palma de Mallorca, Spain; 9Research Institute of Health Sciences (IUNICS), University of Balearic Islands, 07120 Palma de Mallorca, Spain; 10CIBER de Epidemiología y Salud Pública (CIBERESP), Institute of Health Carlos III, 28029 Madrid, Spain; 11Department of Medicine, University of the Balearic Islands, 07122 Palma, Spain; 12Research Group on Nutritional Epidemiology & Cardiovascular Physiopathology (NUTRECOR), Health Research Institute of the Balearic Islands (IdISBa), University Hospital Son Espases (HUSE), 07120 Palma de Mallorca, Spain; 13Statistical and Psychometric Procedures Applied in Health Sciences (PSICOMEST), Health Research Institute of the Balearic Islands (IdISBa), 07120 Palma, Spain; 14Department of Psychology, University of the Balearic Islands (UIB), 07120 Palma, Spain; 15Research Network on Chronicity, Primary Care, and Health Promotion (RICAPPS), Institute of Health Carlos III, 28029 Madrid, Spain; 16Research Group on Global Health and Human Development, University of the Balearic Islands (UIB), 07120 Palma, Spain; 17Son Serra-La Vileta Primary Health Care Center, Balearic Health Service, 07013 Palma, Spain; 18Dra. Teresa Piqué Primary Health Care Center, Balearic Health Service, 07003 Palma, Spain

**Keywords:** prediabetes, mHealth, digital health intervention, SMS intervention, nursing, diet advice, physical activity, prevention, type 2 diabetes mellitus

## Abstract

Background: Type 2 diabetes mellitus (T2DM) is a highly prevalent disease associated with an increased risk of comorbidities, premature death, and health costs. Prediabetes is a stage of glucose alteration previous to T2DM, that can be reversed. The aim of the study is to develop and evaluate a low-intensity, multifaceted, digital intervention to prevent T2DM. The intervention comprises: (1) the use of mobile health technology to send tailored text messages promoting lifestyle changes to people at risk of T2DM and (2) the provision of online education to primary healthcare physicians and nurses about management of prediabetes. Methods: In stages 1–4 we will design, develop and pilot-test the intervention. In Stage 5 we will conduct a phase II, six-month, three-arm, cluster randomized, clinical trial with 42 primary care professionals and 420 patients at risk of T2DM. Patients will be allocated to a control group (usual care), intervention A (patient messaging intervention), or intervention B (patient messaging intervention plus online education to their primary healthcare professionals). The primary outcome will be glycated haemoglobin. All the procedures obtained ethical approval in June 2021 (CEI-IB Ref No: IB4495/21PI). Discussion: Digital health interventions can effectively prevent T2DM and reduce important T2DM risk factors such as overweight or hypertension. In Spain, this type of intervention is understudied. Moreover, there is controversy regarding the type of digital health interventions that are more effective. Findings from this study may contribute to address T2DM prevention, through a low-cost and easily implementable intervention.

## 1. Introduction

Chronic diseases are increasingly prevalent and pose an enormous burden on patients and the healthcare system. Not surprisingly, primary prevention has emerged as a key priority internationally [1].

Recent advances in the area of implementation research show that, even when effective, most interventions frequently fail to be translated into routine healthcare practice due to deficiencies in the implementation strategies. It is estimated that, on average, around 17 years are needed for evidence-based interventions to be integrated into clinical practice, and that almost two-thirds of the organizations’ efforts to implement changes are unsuccessful [2]. Implementation scientists have suggested that nurses could play a key role in streamlining the implementation of healthcare interventions, given their dual role in clinical practice and research [3,4].

The implementation of successful interventions to prevent type 2 diabetes mellitus (T2DM) is particularly challenging. Recommendations for supporting T2DM prevention through healthy lifestyles are now widely incorporated in clinical practice guidelines [5]. However, strategies for providing effective continuing support and motivation are not well developed, and facilitating sustained behaviour change is a key challenge. A number of barriers, both at the patient and at the healthcare professional level, limit the uptake of these interventions. At the patient level, the uptake of lifestyle interventions is influenced not only by intrapersonal participant factors, but also by external determinants related to healthcare professionals or the environment [6]. At the healthcare professional level, their participation in lifestyle-based prevention programs is generally scarce, mainly due to a lack of confidence in the effectiveness of the prevention programs, poor communication skills, lack of time with each patient, lack of specialized staff (nutritionists, sports scientists, psychologists, etc.), lack of training on brief counselling techniques, and limited organizational support [7].

One potential solution to this complex problem could be the use of interventions delivered via mobile devices (mHealth interventions), as a way of offering a new approach to support lifestyle changes [8,9]. Automated messages delivered via such tools can potentially target an extensive range of beliefs and behaviours over a long period of time, being the interventions wide-ranging and low cost. The content of mHealth interventions can also be personalized based on data from electronic health records and patient-generated data, having a great potential for delivering personalized health services. Previous trials showed that brief text messages (e.g., SMS) delivered via digital health systems added to usual care can be effective in improving T2DM risk factors such as overweight [10,11], hypertension [12], hyperlipidemia [13], and high levels of glycated haemoglobin (HbA1c) [10,14]. Several trials have evaluated the impact of mHealth interventions in people at high risk of diabetes, with positive results in reducing T2DM incidence [15], glucose tolerance [16], and weight loss and HbA1c [11]. However, a recent study in India and UK, showed no significant reduction in the progression to T2DM in 2 years by lifestyle modification using SMS messaging [17]. This highlights the need for future studies to analyse the effectiveness of mHealth interventions within other populations, carefully analysing the content of the messages, the behavioural change techniques applied, and the topics that are covered, as well as other characteristics of the intervention, such as the duration or frequency of the messages sent.

Type 2 diabetes mellitus is a common, life-long condition affecting around 4 million people in Spain [18], being one of the largest public health problems worldwide [19]. People with T2DM are at high risk of developing serious complications (e.g., blindness, lower-limb amputations, kidney disease, and cardiovascular disease), which reduce their quality of life and life expectancy. T2DM and its complications are the cause of major expenditure for the Spanish National Health System (SNS): yearly, 8% of total SNS expenditure is spent on direct costs, in addition to other indirect costs of diabetes-related complications. Type 2 diabetes mellitus complications also affect not only the health status of the individual but also their ability to work during their productive life: in Spain, labour productivity losses attributable to T2DM are around €2.8 billion [20]. Furthermore, T2DM and its risk factors disproportionally affect socially vulnerable populations: in Spain, the prevalence of T2DM in the lowest (most deprived) socio-economic status (V to VI) is 9.3%, compared to 4.5% in the upper class (I and II) [21].

Prediabetes, defined by the National Institute for Health and Care Excellence (NICE) [22] and the Working Group of the Spanish Diabetes Society [23] as HbA1c from 6% to 6.4% or fasting plasma glucose 110–125 mg/dL, or both, approximately affects 7.3% of the population worldwide [19]. However, in Spain, the prevalence of prediabetes is higher, around 13–15% [24]. Prediabetes is a high-risk state for diabetes, with an annualized conversion rate of 5–10%; with similar proportion converting back to normoglycaemia [25]. In addition, people with prediabetes have a higher risk of developing cardiovascular diseases [26]. For people with prediabetes, lifestyle modification focusing on a healthy diet alongside the promotion of physical activity, is the cornerstone of diabetes prevention, with evidence of a 40–70% relative-risk reduction [24,25].

Despite the observed large benefits of lifestyle modification, the implementation in the healthcare setting of interventions to support lifestyle changes in people with prediabetes remains an important challenge: patients’ knowledge about diabetes and its prevention is still insufficient among the Spanish population [27], and indicators used to evaluate the efficacy of preventive diabetes interventions have not improved over the last fifteen years [28]. MHealth interventions can reach a wide range of people with a small investment of time and resources, allowing more time to be spent on other healthcare needs. Therefore, it is important to find highly implementable and effective digital interventions to promote healthy lifestyles in people with prediabetes. In Spain, the impact of digital health interventions to promote lifestyle changes in people with prediabetes has not been trialled. Digital health interventions have the potential to be a low cost, highly scalable, and sustainable strategy for Health Systems. Spain is one of the countries with the highest mobile phone use rates in the world, with 98% of users [29]. According to the Spanish National Institute of Statistics latest figures, the proportion of older (>65 years) people who used a mobile phone in the last three months increased from 64% to 89% between 2010 and 2018. Therefore, an intervention based on the use of short, automated messages (SMSs) to promote healthy lifestyles has the potential to reach a very large proportion of the Spanish population. Unlike social media platforms, such as WhatsApp, SMSs have the advantage that information security is guaranteed, does not require Internet access, and it is accessible and usable by socially vulnerable population.

There is an important knowledge gap about the impact of digital interventions to support healthy lifestyles in patients with prediabetes, particularly in Spain. Taking into account the potential for digital interventions to prevent T2DM and its complications, methodologically robust research is needed to develop and evaluate evidence-based digital interventions to support lifestyle change in people at risk of T2DM.

## 2. Materials and Methods

### 2.1. Aims

The main aim is to develop and evaluate a multifaceted digital health intervention to prevent T2DM by supporting lifestyle changes in people at risk of T2DM based on: (1) the use of a system comprising mHealth technology to send automated, tailored brief text messages and (2) the provision of education to primary healthcare professionals about T2DM prevention and management of prediabetes.

The secondary aims are:To develop a multifaceted, digital health intervention to prevent T2DM.To pilot-test and optimize the components of a digital health intervention.To explore the effects of the digital health intervention on glycated haemoglobin (primary outcome) and on additional clinical, physiological, behavioural and psychological outcomes through a phase II, 3-arm, 6-month clinical trial.To test the feasibility of a future full-scale phase III clinical trial, quantifying the number of eligible patients, recruitment rate, and follow-up rate.

### 2.2. Hypothesis

It is feasible to develop a multifaceted, digital health intervention to prevent T2DM based on: (1) the use of a system comprising mobile health technology integrated with electronic health records to send automated, tailored brief text messages supporting lifestyle changes in people at risk of T2DM and (2) the provision of online education to primary healthcare professionals about T2DM prevention.The proposed interventions are feasible to deliver and acceptable to patients and primary healthcare professionals.Compared to the control group, the proposed interventions reduce HbA1c (primary outcome) at least 0.3% and improve additional clinical, physiological, behavioural and psychological outcomes.

### 2.3. Design

The process for the development of the PREDIABETEXT intervention is based on the MRC Guidance for the Development of Complex Interventions [30]. This study involves five stages (Figure 1). Each stage consists of various tasks divided into sub-stages. The first 4 stages consist of the development and pilot-testing and refinement of the interventions, whereas stage 5 will involve a phase II trial with an embedded qualitative study to evaluate the interventions and trial procedures.

#### 2.3.1. STAGE 1: Developing the Text Messaging Intervention

Stage 1 aims to design the SMS library and will involve three sub-stages:

Sub-stage 1.1. Exploring patient perspectives about brief messages supporting lifestyle change behaviour: interviews with 15 patients (diverse in terms of socioeconomic level) at risk of T2DM (HbA1c 6–6.4% over the last three months, two consecutive values of fasting plasma glucose 110–125 mg/dL over the last 12 months, or both) will be conducted to understand the way messages might work in supporting lifestyle changes. Interviews will be audio-recorded, transcribed and analysed using the thematic analysis technique [31,32].

Sub-stage 1.2. Exploring healthcare professionals’ perceived barriers and facilitators to intervention implementation: interviews with 15 primary healthcare general practitioners (GP) and nurses will be conducted to identify barriers to the implementation of the intervention. The interviews will be analysed using thematic analysis [32].

Sub-stage 1.3. Producing the library of messages: The SMS library created for the DIABE-TEXT study [33] will be used as a starting point and the messages will be adapted for the pre-diabetes population. The DIABE-TEXT library was created from an expert workshop with endocrinologists, nutritionists, sports scientists, nurses, psychologists, GPs and pharmacists, which generated up to 500 messages using different behaviour change techniques [34]. A team of nutritionists and sports scientists will adapt the messages from DIABE-TEXT study and create new messages with the aim of having messages covering all the relevant topics (motivation, type of food, portions, physical activity, sedentary lifestyle, etc.) adapted for the pre-diabetes population. Once the new SMS library has been created, other professionals will review the messages and make suggestions for improvement. Once we have the final SMS library, we will move on to the design of the delivery strategy. A team of nutritionists and sports scientists will arrange the messages in the order in which they will be sent to patients, taking into account the variety of messages (motivation, pre-diabetic education, nutrition, physical activity and sedentary lifestyle), progression in complexity, progression in intensity, frequency and volume in the case of physical activity, and personalisation (smoking and weight loss).

The message library will then be externally reviewed by three patients that will have participated in the previous qualitative study (sub-stage 1.1) giving constructive feedback to ensure that the content is appropriate and understandable. These patients will read the complete library and identify those messages that do not comply with these characteristics. In the same way, two diabetes experts will also review the library, to ensure comprehensiveness and alignment with current guidelines. The library will be then modified if necessary.

#### 2.3.2. STAGE 2: Adapting Our Existing Technology Systems to Deliver Text Messages

As part of a nationally funded ongoing project, members of our team have already created a technological system that allows sending personalized text messages based on available data from electronic health records. This system has three main components: (1) a database with selected data from the study participants extracted from the PRISIB (the IdISBa research platform on health information), (2) the message library, and (3) the platform to send SMS, hosted by the Balearic Islands Health System (Figure 2). In stage 2, this system will be adapted to the PREDIABETEXT study by developing new algorithms and programming the creation of a new bespoke clinical database.

#### 2.3.3. STAGE 3: Developing an Online Educational Intervention Targeted at Primary Healthcare Professionals

This intervention will aim to (1) raise awareness about the importance of T2DM prevention, (2) increase knowledge about effective strategies for diabetes prevention, (3) improve knowledge about brief counselling techniques, and (4) enhance communication skills. The development of this intervention will involve three sub-stages:

Sub-stage 3.1. Exploring healthcare professionals’ educational intervention needs and preferences and perceived barriers and facilitators to its implementation: interviews with 15 primary healthcare GP and nurses, working in several settings and having different knowledge of prediabetes to ensure diversity, will be conducted to inform the design of the educational intervention. They will also be asked about the perceived barriers and facilitators for implementing the educational intervention among the healthcare professionals’ community. The interviews will be analysed following the thematic analysis technique [32].

Sub-stage 3.2. Development of the content of the educational intervention: This educational intervention will consist of a complete course about the management of patients with prediabetes at the primary healthcare level. Content blocks such as epidemiology, treatment, complications or motivation will be addressed—with the support of different activities—including solving clinical cases and watching clinical practice simulation videos.

Sub-stage 3.3. Digitalization of the intervention: the materials will be uploaded to the Moodle online platform from the Mallorca Primary Healthcare Teaching Unit.

#### 2.3.4. STAGE 4: Optimizing the Interventions

To test the intervention and make the necessary improvements before carrying out the phase II study, a pre-pilot study will be conducted involving 30 patients at risk of T2DM, and 10 Primary healthcare GPs and nurses. This will involve the following sub-stages:

Sub-stage 4.1. Optimising the messaging intervention: We will recruit 30 people with prediabetes, ensuring representation from all socioeconomic levels [35], to examine the acceptability, relevance, and perceived impact of the brief messages. They will use the messaging system for one month. We will conduct individual semi-structured interviews with 10 patients diverse in terms of socioeconomic level to explore their experience of using the system. The results will be used to further refine the intervention (message content, frequency, level of personalization, subject of the message, etc.).

Sub-stage 4.2. Optimising the educational online intervention: Ten GPs and nurses will be recruited. They will receive the educational intervention and complete an ad-hoc questionnaire about its usefulness. The intervention will then be refined based on their feedback.

#### 2.3.5. STAGE 5: Phase II Trial to Explore Interventions Efficacy and Identify Implementation Barriers

Stage 5 of the study will consist of a Phase II, six-month, three-arm, cluster randomized, clinical trial (Figure 3 and Figure 4). The protocol of this phase II trial was registered on clinicaltrials.gov (NCT05110625) (**Protocol version: v1.1, 16/08/2022**). Table 1 summarizes all items included in the trial registry, as suggested by the World Health Organization [36].

In the Intervention A group, patients will receive the text messaging intervention. In the intervention B group, patients will receive the text messaging intervention and their primary healthcare professionals will receive the online education intervention. Control group participants will receive usual care, and their healthcare professionals will not receive the online educational intervention.

### 2.4. Participants

#### 2.4.1. Primary Healthcare Professionals

Selection criteria: We will include GPs and nurses from primary care centres in Mallorca that agree to participate. We will exclude those planning to move to a different centre during the study period.

Recruitment procedure: We will invite GPs and nurses by email (containing all the relevant information about the study) and a phone call to confirm participation. We will provide participant information sheets and will obtain informed consent (audio recorded over the phone). We will prioritize the recruitment of those primary care professionals with a higher number of patients meeting criteria for prediabetes (detailed below).

#### 2.4.2. Patients

Selection criteria: We will include patients between 18–75 years old, registered in the Public Health Service of the Balearic Islands, at risk of T2DM (HbA1c from 6% to 6.4% in the last three months, two consecutive values of fasting plasma glucose between 110–125 mg/dL, or both) [23], with access to a mobile device able to receive text messages. We will exclude those not able to read messages in Spanish or with severe mental conditions. Patients taking antidiabetic drugs will also be excluded as well as women who gave birth in the previous 12 months or that are currently pregnant and people planning to change healthcare centre during the intervention period.

Recruitment procedure: With support from the PRISIB platform (health information platform of the Health Research Institute of the Balearic Islands), we will extract from electronic health records a list of all patients from the health professionals previously recruited with a record during the last 12 months of: (1) HbA1c from 6–6.4% or (2) two consecutive values of fasting plasma glucose between 110–125 mg/dL or (3) both. Starting from those patients with most recent blood test determinations available, we will send to patients an SMS inviting them to participate in the study. This SMS will include a link to a website containing information about the study (brief summary, patient information sheet, and informed consent). Within the next 48 h, a research assistant will phone them to formally invite them to participate in the study, provide additional information (if needed), and obtain the informed consent (audio-recorded over the phone). Blood tests will be ordered for all patients with test results recorded more than three months ago from the recruitment date to confirm patient eligibility.

### 2.5. Sample Size Determination

The sample size was estimated based on HbA1c as the main outcome variable. We plan to recruit 42 primary healthcare nurses or GPs and 420 patients (10 patients per provider). Assuming an 80% power, an intraclass correlation coefficient of 0.04 [37], a design effect of 1.36, and a dropout rate of 10%, our sample-size calculation indicates that 420 patients (control group: 140; Intervention group A: 140; Intervention group B: 140) would allow us to detect a difference in HbA1c of at least 0.3% between the control and the intervention groups (assuming standard deviation = 0.8%). According to previous research [38], this difference would allow us to detect statistically significant differences in diabetes incidence in a future phase III trial with a longer (at least 12 months) follow-up.

### 2.6. Randomization and Masking

Family physicians or nurse participants in the study will be randomized by computer-generated random numbers to one of the three arms of the study. Patients from each physician’s or nurse’s will be invited to participate and will receive the intervention according to their physician or nurse arm allocation: Arm 1: healthcare professionals will receive online training and their patients will receive SMS; Arm 2: their patients will receive SMS; Arm 3 (control group): usual care. Outcome assessors will be blinded to treatment allocation.

### 2.7. Interventions

#### Description of the Intervention Group (SMSs)

In the Intervention A group, patients will receive the text messaging intervention, with a frequency of 3 to 5 messages per week, depending on the results of the pre-pilot stage (project stage 4), where the perception of the participants will be assessed, both in terms of frequency and content of the SMS. Messages will be received from Monday to Friday, never on weekends or holidays because of technological limitations of the system available to be used. The structure of the messages will be designed to vary the subject matter (motivation, type of food, portions, physical activity, sedentary lifestyle, etc.); as well as to follow a progression in the complexity of the messages and the physical activity recommendations.

The messages will focus on nutrition and physical activity. In terms of nutrition, recommendations will be given on the type of carbohydrates that are most suitable to prevent glycaemic peaks, portions, and frequency of consumption of food groups, ultra-processed foods and others for occasional consumption, alcohol consumption and energy drinks, as well as recipe examples [39,40,41]. In terms of physical activity, messages will include recommendations to increase the frequency, intensity and duration of physical activity, strategies to reduce sedentary behaviour, and strategies to replace sedentary time with physical activity time [42,43]. In addition, we will include motivational messages, as well as other personalized messages for participants who need to lose weight or who are smokers.

In the intervention B group, patients will receive the same text messaging intervention as in group A, and in addition, their primary healthcare professionals will receive online education on the management of patients with prediabetes at the primary healthcare level. The online education for healthcare professionals will consist of written material on how to achieve behavioural changes and what recommendations are appropriate for a population with prediabetes in terms of diet, exercise and diabetes prevention. They will also be provided with video examples of clinical consultations. These interviews have been designed by a group of experts, including sports scientists, dietitian-nutritionists, GPs, psychologists, and nurses.

Control group participants will receive usual care, and their healthcare professionals will not receive online education.

### 2.8. Participant Timeline

Table 2 provides an overview of the time schedule of enrolment, interventions, main study visits and assessment measures for participants. Each of the visits is explained in detail below.

Visit −3 (day −60/−45): Invitation to participate. Potential eligible primary healthcare professionals will receive the invitation by email. Once they agree to participate and sign the informed consent form, will be randomized. Potentially eligible participants will receive the invitation by SMS.

Visit −2 (day −45/−7): Participants’ recruitment and baseline interview. The research assistant will contact potential participants via phone. On this call, patients will be given thorough information on the study and, if willing to participate, they will be asked for informed consent. On the same call, a basal interview (behavioural and psychological outcomes) will be performed.

Visit −1 (day −7/−1): Baseline data collection. Participants who signed the informed consent will undergo baseline data collection at the hospital (anthropometric measurements, blood pressure and blood laboratory examinations).

Visit 0 (day 0): Initiation of the intervention. Patients in the Intervention A and Intervention B group will receive their first SMS. Patients in the control group will be informed of their allocation. Health professionals who have been assigned to the intervention B group will take the online education. They will have 4 weeks to complete it.

Visit 1 (month 6): Post intervention data collection. After 6 months of intervention, data collection from patients will be performed following the same procedure as at baseline time basal interview. Interview on behavioural and psychological outcomes will be conducted by phone, and anthropometric measurements, blood pressure and blood laboratory examinations at the Clinical trial Platform at IdISBa. Data collection from health professionals will also be accomplished during this time, through individual interviews (only with Intervention B group professionals) and by delivering an online questionnaire.

### 2.9. Data Collection

We will collect specific data from participating patients and healthcare professionals.

At the patient level, baseline data will be collected over the phone immediately after recruitment with a structured interview using the Lime Survey online tool. Patients will subsequently be invited to attend a visit at the IdISBa Clinical Trial Platform, for the collection of basal clinical and physiological data by a research nurse. Post-intervention data will be collected after 6 months in the same way as in baseline data collection. We will also examine the proportion of participants exposed to the intervention (i.e., successfully receiving messages) at 6-months follow-up and conduct around 15 individual interviews to explore patient experiences with the intervention.

At the professional level, data collection will be conducted using structured questionnaires to (1) examine the proportion of primary healthcare professionals actively adopting the intervention program and (2) assess the extent to which they improved their clinical management of prediabetes. We will also conduct around 10 individual interviews with primary healthcare professionals with high or low implementation ratios to identify barriers to intervention delivery and implementation.

### 2.10. Outcome Measures

The primary outcome measure will be glycated haemoglobin (HbA1c) at 6 months follow-up. Secondary outcome measures will include:

(A) Clinical and physiological outcomes: Fasting blood glucose, triglycerides, total cholesterol, low density lipoprotein cholesterol (LDL), high density lipoprotein cholesterol (HDL), lipoprotein A, aspartate aminotransferase enzyme (AST), alanine aminotransferase (ALT), gamma glutamyl transpeptidase (GGT), complete hemogram, creatinine, serum albumin, insulin, urinary sediment, albumin-creatinine ratio, body weight, waist circumference, blood pressure, and cardiovascular disease risk [44].

(B) Diabetes incidence: proportion of patients with a new diagnosis of T2DM.

(C) Motivational and behavioural outcomes: brief motivation questionnaire (ad hoc); adherence to the Mediterranean diet, measured with MEDAS questionnaire [45]; physical activity, measured with REGICOR Short form Physical Activity Questionnaire [46]; sedentary behaviour, measured with the Spanish version of the physical activity questionnaire used in the Physical activity questionnaire from the Nurses’ Health Study [47]; smoking habit, measured with an adapted version of the Global Adult Tobacco Survey: GATS questionnaire; and alcohol consumption will be collected.

(D) Trial feasibility outcomes: numbers of eligible patients, recruitment rate, follow-up rate.

### 2.11. Data Analysis

We will use descriptive statistics to explore the characteristics of the participants at baseline. To estimate the presence and magnitude of a potential selection bias, we will compare the characteristics of healthcare professionals and patients recruited vs the characteristics of the complete population of healthcare professionals and patients meeting our eligibility criteria (defined above). The impact of the intervention will be measured in terms of the difference between groups in the primary and secondary outcomes after the 6-month intervention period. For that end, we will build general linear models (ANCOVA), adjusting for baseline values. All the analyses will be conducted by the intention to treat (ITT) approach. Missing values will be replaced using the multiple imputation model (MICE) [48].

Statistical analysis will be performed using SPSS v.25 (IBM, New York, NY, USA) and Stata v.16 (Statacorp, College Station, TX, USA), and we will use an α of 5% throughout.

### 2.12. Ethical Considerations

The study complies with the ethical standards of the Declaration of Helsinki and all procedures were approved in June 2021 by the Institutional Research Ethics Review Board of the Balearic Islands Health Service (CEI-IB Ref No: IB4495/21PI). Participants’ consent will be obtained, and all data collected will be kept confidential and anonymous. As specified in the informed consent, in case of withdrawal, all data collected prior to patient withdrawal will be retained and used by the study investigators. Participants can voluntarily withdraw their informed consent at any time during the study.

### 2.13. Validity and Reliability

This study may generate new evidence about the impact of mHealth interventions in the area of T2DM prevention. The results of the trial are likely to offer a strong internal validity, as we will follow a randomization process based on allocation sequence generation, blinded to the personal investigator and research staff involved in the study. There will be no potential for recruitment bias because randomization will occur after actual participants are enrolled. A pilot study will be conducted during the initial phase of the trial to assess the appropriateness of intervention messages and determine the feasibility of logistic procedures. The study interventions are based on official guidelines and the latest scientific evidence. Data collected with questionnaires might be subjected to potential reporting biases. Nevertheless, this study will mainly use validated questionnaires with robust psychometric properties. Moreover, objective measures (biochemical and anthropometric parameters) will be collected, including the primary outcome HbA1c.

## 3. Discussion

In Spain, more than 4 million people suffer from T2DM. It is well established that T2DM can be effectively prevented through lifestyle changes, but patients at risk of T2DM do not always receive adequate advice on recommended changes in diet and physical activity. Moreover, strategies to provide ongoing and effective support and motivation are not well developed and facilitating sustained behaviour change remains a major challenge.

The interventions proposed in this study have the potential to contribute to addressing this important problem. Two systematic reviews by members of our team show that automated brief text messages sent to mobile devices effectively promote lifestyle changes [14] and medication adherence [49]. We propose these types of interventions not only because they are likely to be effective but also because of their high translational value. They can be delivered at low cost (clear potential for cost-effectiveness) and easily implemented widely (high scalability), therefore constituting a sustainable strategy for Health Services to improve health at the population level. This technology-based solution is highly transferable as it could also be adapted for its use in other settings and groups of patients. In addition, the intervention would offer, in times of pandemic, a feasible alternative to face-to-face consultations for people with prediabetes, with a personalized lifestyle change, using preventive strategies.

These interventions would enable personalized remote health care for a broad population including socially vulnerable populations. This is of particular relevance in view of the affection of T2DM in this population group and the difficulties of access, and therefore representation, of this population group in clinical trials.

Telemedicine has increased during the COVID-19 pandemic, becoming a very important resource for improving healthcare management, especially in primary care, supervised by multidisciplinary teams, consisting mainly of nurses and GPs [50]. Implementing mHealth interventions by nurses for the prevention of T2DM could reduce healthcare costs and reach larger populations.

### Limitations

The study has several limitations. First, due to the nature of the intervention, participants will be aware of their treatment allocation. Second, involving primary healthcare professionals through online intervention may be unfeasible if the COVID-19 pandemic continues hitting our region with high intensity. Finally, the proposed phase II trial is not powered to determine the effectiveness of the proposed interventions in preventing T2DM. However, it will provide valuable information to determine the feasibility of the trial procedures and to explore the potential impact of the intervention in terms of reduction of HbA1c. We plan to conduct a fully powered phase III trial in case positive results are obtained from this study.

## 4. Conclusions

The realization of the present project is key for several reasons: (1) The incidence of type 2 diabetes and pre-diabetes is increasing, and consequently, the associated comorbidities, loss of quality of life, and loss of autonomy; (2) The direct health costs and indirect costs for type 2 diabetes treatment, as labour cost due to lack of productivity, are an important economic burden for many countries. Interventions to prevent the development of type 2 diabetes are needed; (3) The development of a low-cost, easy-to-implement intervention is crucial to address a problem that affects a large vast of population. For all the aforementioned, findings from this study will contribute to solving a real and urgent problem of society, contributing to addressing type 2 diabetes mellitus prevention through a low-cost and easily implementable intervention.

## Figures and Tables

**Figure 1 ijerph-19-14706-f001:**
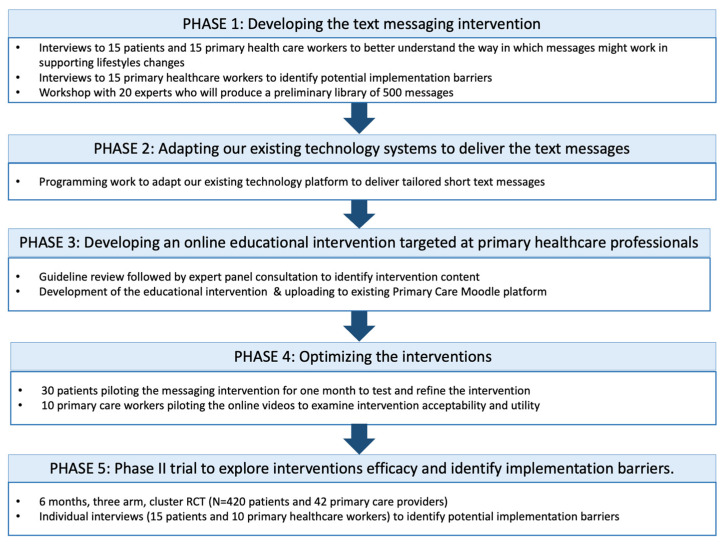
PreDiabeText project overview.

**Figure 2 ijerph-19-14706-f002:**
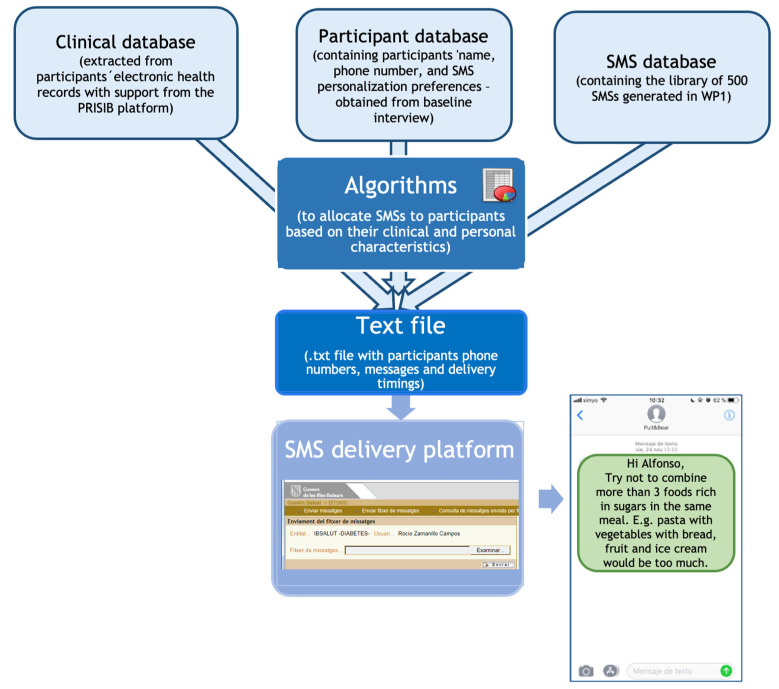
Description of the PreDiabeText Technological System for the messaging intervention. Three databases are combined using specific algorithms in order to produce a text file containing the information to be uploaded to the SMS delivery platform, which will send personalized SMSs to the study participants.

**Figure 3 ijerph-19-14706-f003:**
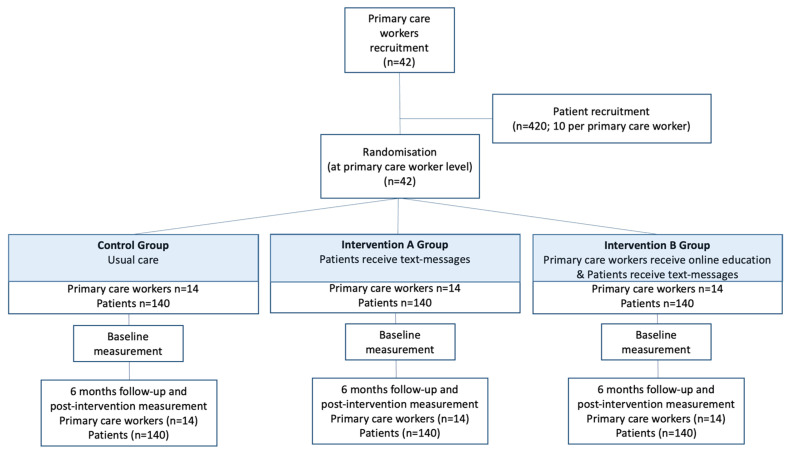
Consort flowchart of the PreDiabeText phase II clinical trial.

**Figure 4 ijerph-19-14706-f004:**
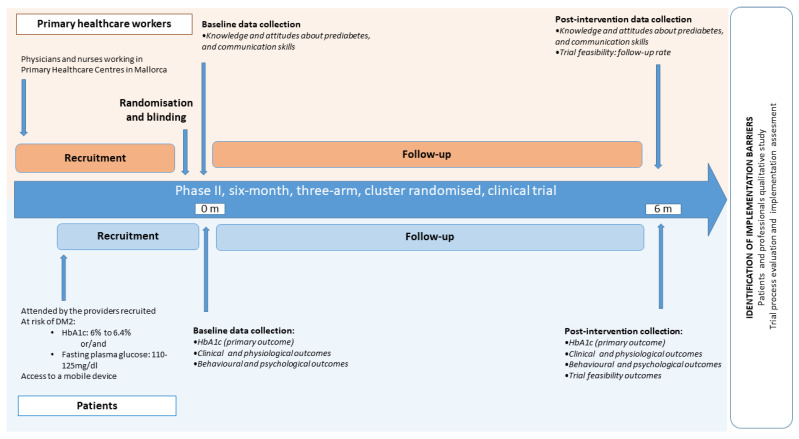
Phase II PreDiabeText study timeline.

**Table 1 ijerph-19-14706-t001:** Brief structured summary of the trial.

Data Category	Information
Trial identification number	ClinicalTrials.gov NCT05110625
Date of registration	8 November 2021
Sponsor	Balearic Islands Research Institute (IdISBa), Spain
Contact for public and scientific queries	Dr Ignacio Ricci Cabello. Primary Care Research Unit of Mallorca (IB-Salut), Balearic Health Service, Palma de Mallorca, Spain, Palma de Mallorca, ignacio.ricci@ibsalut.es
Scientific title	Effects of a low intensity, multifaceted, mHealth intervention to prevent type 2 diabetes mellitus in adults with prediabetes in the primary care setting (the PREDIABETEXT trial)
Country of recruitment	Spain
Health condition studies	Prediabetes
Interventions	Intervention A: Participants will receive text messages (three per week) in their mobile phones during six months Intervention B: Participants will receive messaging intervention plus online education to their primary healthcare workers
	Control: Participants will receive usual care only, and their healthcare workers will not receive online education
Inclusion/exclusion criteria	Eligible age: 15–75 years; eligible sex: males and females.
	Inclusion criteria: Registered in the Public Health Service of the Balearic Islands. HbA1c from 6% to 6.4% or fasting plasma glucose 110–125 mg/dL, or both. With access to a mobile device able to receive text messages.
	Exclusion criteria: documented history of T2D and/or use of oral antidiabetic medication. Younger than 18 years old or older than 75 years old. People not able to read messages in Spanish. Patients with severe mental conditions.
Study type	Interventional
	Allocation: randomized; Intervention model: parallel assignment; masking: none (open label)
	Primary purpose: Change from Baseline HbA1C at 6 months
Date of first enrolment	1 September 2021
Target sample size	420 (140 participants in each group)
Recruitment status	Actively recruiting
Primary outcomes	Reduction of HbA1C
Secondary outcomes	Fasting blood glucose, body weight, waist circumference, blood pressure, lipids, cardiovascular disease risk, proportion of patients developing T2DM, adherence to Mediterranean diet, physical activity, sedentary behaviour, smoking habit, alcohol consumption

**Table 2 ijerph-19-14706-t002:** Participants main visits and assessment schedules.

Main Visits and Assessment Schedules					
Visit	V −3	V −2	V −1	V 0	V 1
Time point ^1^	−60 d	−45 d	−7 d	0 d	6 m
**Participants—patients**					
Invitation by SMS	X				
Informed consent		X			
Inclusion/exclusion criteria		X			
SB assessment ^2^		X			X
Dietary assessment ^3^		X			X
Motivation Questionnaire (ad hoc) ^4^		X			X
PA assessment ^5^		X			X
Randomization ^6^			X		
Blood laboratory examinations ^7^			X		X
Anthropometric measurements ^8^			X		X
Blood pressure measurement ^9^			X		X
Initiation of the intervention ^10^				X	
Trial feasibility: follow-up rate					X
**Participants—health care workers**					
Invitation by email	X				
Informed consent	X				
Inclusion/exclusion criteria	X				
Randomization	X				
Online education (intervention B group)				X	
Individual interviews (intervention B group)					X
Interview: Knowledge and attitudes about prediabetes, and communication skills					X
Trial feasibility: follow-up rate					X

1. Time point: d = day; m = month; 2. SB assessment: NHS questionnaire; 3. Dietary assessment: MedDiet adherence questionnaire; 4. Motivation Questionnaire (ad hoc); 5. PA assessment: REGICOR PA questionnaire; 6. Randomization: Patients and healthcare workers will be randomised at the same time, with only the healthcare worker being randomised, and their 10 patients being assigned to the same intervention group as the healthcare workers; 7. Blood laboratory examinations: glucose, HbA1C, total cholesterol, HDL-c, LDL-c, TG, Hb, MCV, serum creatinine, AST, ALT, and GGT. TyG index and FLI will be calculated; 8. Anthropometric measurements: body weight, WC and HC, BMI, body composition, height; 9. Blood pressure measurement: the arm with the highest blood pressure. Subsequent blood pressure measurements will be taken in this arm throughout the study; 10. Initiation of intervention: initiation of SMS sending.

## Data Availability

Not applicable.

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
