# Peer review of "Development and Evaluation of a Digital Health Intervention to Prevent Type 2 Diabetes in Primary Care: The PREDIABETEXT Study Protocol for a Randomised Clinical Trial"

_ijerph, 2022, doi:10.3390/ijerph192214706_

Round 1
Reviewer 1 Report
The project is interesting and can provide useful information on the subject they develop. In my opinion, the authors should explain in greater detail how they are going to randomly recruit the participants in the different phases using the PRISIB platform or others, also taking into account representation by gender, ages and age groups, to avoid research biases.
Author Response
Reviewer #1
Comment #1: The project is interesting and can provide useful information on the subject they develop. In my opinion, the authors should explain in greater detail how they are going to randomly recruit the participants in the different phases using the PRISIB platform or others, also taking into account representation by gender, age and age groups, to avoid research biases.
Response to Comment #1: Thank you for reviewing our manuscript and for the positive feedback provided. We are not planning to recruit a random sample of participants. Instead, for the recruitment of healthcare professionals, we will prioritize those professionals with a higher number of patients meeting criteria for prediabetes (we will rank healthcare professionals according to the number of patients meeting criteria for prediabetes and will start inviting those with a higher number of patients). Similarly, for the recruitment of patients, we will prioritize those patients with the most recent laboratory test undertaken. This has now been clarified as follows:
“Recruitment procedure [Primary health care professionals]: we will invite t GPs and nurses by email (containing all the relevant information about the study), and a phone call to confirm participation. We will provide participant information sheets and will obtain informed consent (audio recorded over the phone). We will prioritize the recruitment of those primary care professionals with a higher number of patients meeting criteria for prediabetes (detailed below).” (Page 10)
“Recruitment procedure [Patients]: with support from the PRISIB platform (health information platform of the Health Research Institute of the Balearic Islands), we will extract from electronic health records a list of all patients from the health professionals previously recruited with a record during the last 12 months of: i) HbA1c from 6%-6.4%, or; ii) two consecutive values of fasting plasma glucose between 110-125 mg/dl, or; iii) both. Starting from those patients with most recent blood test determinations available, we will send to patients an SMS inviting them to participate in the study. This SMS will include a link to a website containing information about the study (brief summary, patient Information sheet, and informed consent). Within the next 48 hours, a research assistant will phone them to formally invite them to participate in the study, providing additional information (if needed), and obtaining the informed consent (audio-recorded over the phone). Blood tests will be ordered for all patients with test results recorded more than three months ago from the recruitment date to confirm patient eligibility.” (Page 10)
We now included, as part of the “Data Analysis” section, additional details about how we will estimate the existence of a potential selection bias: “To estimate the presence and magnitude of a potential selection bias, we will compare the characteristics of healthcare professionals and patients recruited vs the characteristics of the complete population of healthcare professionals and patients meeting our eligibility criteria (defined above).” (Page 13)
Reviewer 2 Report
General Comments:
Major clinical trials, including the Harvard Study by Khera AV and associates, NEJM 375, 2349-2358, 2016 have demonstrated the beneficial effects of lifestyle changes on lowering premature mortality due to acute vascular events. Therefore, it is worthwhile studying such interventions in cost-effective ways.
The authors have proposed the development of a digital approach to the management of type-2 diabetes. It is just a proposal, and not research study or the results of such a field study. If it works, it is relevant. We will know only after they develop and implement the study. Since it is supposed to be a new approach, and has not been tested, it is hard to compare it to existing approaches, which are not very exhaustive. The proposal is well written and is easy to understand. Since studies have not been done, it is hard to predict the validity of any conclusions one may develop in such a proposal.
Specific Comments:
As a research proposal, it is quite through, unique, and well written. If they develop and implement the proposal, it is possible that the data generated will provide new and useful information for the management of type-2 diabetes. Having said that, it is worth remembering, that in any such public health project, success depends on patient compliance. How do you follow the compliance? How do you account for failure of patient compliance? The real challenge is patient adherence of recommendations, and the patient experience matters. Hence patient communication, unbiased, evidence-based recommendations and evaluation of adherence becomes very important. Questions like these and many others, will arise only after the proposal is implemented. At this point, these questions are just speculations.
Author Response
Reviewer #2
General Comments:
Major clinical trials, including the Harvard Study by Khera AV and associates, NEJM 375, 2349-2358, 2016 have demonstrated the beneficial effects of lifestyle changes on lowering premature mortality due to acute vascular events. Therefore, it is worthwhile studying such interventions in cost-effective ways.
The authors have proposed the development of a digital approach to the management of type-2 diabetes. It is just a proposal, and not research study or the results of such a field study. If it works, it is relevant. We will know only after they develop and implement the study. Since it is supposed to be a new approach, and has not been tested, it is hard to compare it to existing approaches, which are not very exhaustive. The proposal is well written and is easy to understand. Since studies have not been done, it is hard to predict the validity of any conclusions one may develop in such a proposal.
Response to General Comment:
Thank you for reviewing our manuscript and for considering our research worthwhile and relevant.
Specific Comments:
As a research proposal, it is quite through, unique, and well written. If they develop and implement the proposal, it is possible that the data generated will provide new and useful information for the management of type-2 diabetes. Having said that, it is worth remembering, that in any such public health project, success depends on patient compliance. How do you follow the compliance? How do you account for failure of patient compliance? The real challenge is patient adherence of recommendations, and the patient experience matters. Hence patient communication, unbiased, evidence-based recommendations and evaluation of adherence becomes very important. Questions like these and many others, will arise only after the proposal is implemented. At this point, these questions are just speculations.
Response to Specific Comment:
We fully agree with Reviewer #2 that it is necessary to be able to carry out the study to answer these questions, and undoubtedly a very important part of the effectiveness of interventions on lifestyle changes in humans depends on adherence to the intervention. For this reason, a multidisciplinary team (including health psychologists) will design the messaging intervention, applying well-established behavioural change techniques. This has been detailed in the manuscript as follows:
“The DIABE-TEXT library was created from an expert workshop with endocrinologists, nutritionists, sports scientists, nurses, psychologists, GPs and pharmacists, which generated up to 500 messages using different behaviour change techniques [34].” (Page 5)
Actual adherence to the recommendations for the prevention of type 2 diabetes provided as part of the intervention will be assessed based on patient-reported data, using psychometrically robust questionnaires. This information is detailed in the manuscript as follows:
“C) Motivational and behavioural outcomes: brief motivation questionnaire (ad hoc); adherence to the Mediterranean diet, measured with MEDAS questionnaire [45]; physical activity, measured with REGICOR Short form Physical Activity Questionnaire [46]; sedentary behaviour, measured with the Spanish version of the physical activity questionnaire used in the Physical activity questionnaire from the Nurses' Health Study [47]; smoking habit, measured with an adapted version of the Global Adult Tobacco Survey: GATS questionnaire; and alcohol consumption will be collected.” (Page 13)
We also agree with Reviewer 2 about the importance of taking into account the patient experience. As a matter of fact, we will use qualitative research methods to collect and analyse data from participants in the intervention groups:
“We will (…) conduct around 15 individual interviews to explore patient experiences with the intervention.” (Page 12)
Round 2
Reviewer 1 Report
The authors have clarified the doubts raised and have better drafted the inclusion criteria.